# Relationship between the Social Development Index and Self-Reported Periodontal Conditions

**DOI:** 10.3390/healthcare11111548

**Published:** 2023-05-25

**Authors:** Mireya Martínez-García, Adriana-Patricia Rodríguez-Hernández, Guadalupe O. Gutiérrez-Esparza, Roberto Carlos Castrejón-Pérez, Enrique Hernández-Lemus, Socorro Aída Borges-Yáñez

**Affiliations:** 1Department of Immunology, Instituto Nacional de Cardiología Ignacio Chávez, Mexico City 14080, Mexico; 2Laboratory of Molecular Genetics, Graduate Studies and Research Division, School of Dentistry, Universidad Nacional Autónoma de México, Mexico City 04510, Mexico; aprh_gm@fo.odonto.unam.mx; 3Cátedras CONACYT Consejo Nacional de Ciencia y Tecnología, Mexico City 08400, Mexico; ggutierreze@conacyt.mx; 4Instituto Nacional de Cardiología Ignacio Chávez, Mexico City 14080, Mexico; 5Department of Clinical and Epidemiological Geriatric Research, Instituto Nacional de Geriatría, Mexico City 10200, Mexico; rc.castrejon.perez@gmail.com; 6Computational Genomics Division, Instituto Nacional de Medicina Genómica, Mexico City 14610, Mexico; ehernandez@inmegen.gob.mx; 7Center for Complexity Sciences, Universidad Nacional Autónoma de México, Mexico City 04510, Mexico; 8Dental Public Health Department, Graduate Studies and Research Division, School of Dentistry, Universidad Nacional Autónoma de México, Mexico City 04510, Mexico

**Keywords:** periodontal disease, bone loss, oral health surveys, Social Development Index, living conditions, oral public health

## Abstract

Inequalities in oral health are influenced by the social strata of the population. Few studies have focused on the multitude of factors related to social development as indicators of living conditions and periodontal health status. The aim of this study is to evaluate the association between self-reported periodontal conditions and the Social Development Index (SDI). A cross-sectional validated questionnaire was carried out among 1294 Mexican adults. Descriptive statistics and multivariate logistic regression models were used to identify the best predictors of self-reported periodontal conditions. Bone loss reporting was used as a proxy for the presence of periodontal disease. We found that higher global scores on the SDI and quality and available space in the home (QASH) increase the probability of having bone loss. Global SDI (OR = 7.27) and higher QASH (OR = 3.66) were indeed the leading societal factors related to periodontal disease. These results have pointed out how SDI and its indicators, in particular QASH, can be used to further explore inequities related to privileged access to dental care in the context of periodontal diseases.

## 1. Introduction

Periodontal diseases (PD) are considered a major silent public health burden globally [1]. PD can indeed exist for a long time before causing disability and pain [2,3,4]. The most recently reported prevalence ranges from 20% to 50% worldwide [5]. Gingivitis, the mildest form of PD, is characterized by bleeding, bad breath, and swollen gums and can be found in up to 90% of the population [6]. Periodontitis is a stage of disease progression beyond gingivitis, characterized by the disruption of biofilm ecology and the irreversible destruction of the periodontium. It is a chronic and inflammatory pathology [7,8]. Currently, periodontitis is considered a challenging public health problem due to its high prevalence. It is the most common non-communicable chronic inflammatory disease, as well as a source of social inequality that has a negative impact on quality of life and on general health, and it is associated with significant medical and dental care costs [9].

The prevalence and severity of PD gradually increase with age, peaking in 40-year-olds. Common risk factors include genetic variations, systemic diseases, and an association with SARS-CoV-2 infection, which was recently reported [10,11,12,13,14,15,16,17]. Several environmental and modifiable factors have also been associated with the different manifestations of periodontal diseases and dysbiosis of subgingival microbiota [11,12,13,14,15,16]. Latin American populations, in particular, are significantly affected by these complex diseases [18]. Demographic and socioeconomic factors have been clearly associated with periodontal pathologies [19].

Mexico is one of the largest countries in terms of population and economy in Latin America. However, it is also one of the countries that faces significant socioeconomic disparities, resulting in vulnerabilities and social health inequalities [19,20]. This disparity is reflected in the minimal public funding of healthcare systems, particularly in oral health care. As a result, the Mexican population is forced to finance dental care in the private sector, and specialized treatments such as periodontal care are often indefinitely postponed due to financial constraints [21,22]. In recent years, it has been estimated that between 50% and 70% of the Mexican population is affected by some form of periodontal disease [23,24].

Few studies have been carried out regarding the association of social security, sociodemographic context, and socioeconomic strata variables with oral health conditions in the Mexican population [25,26,27]. However, other aspects related to social development, such as the Gini coefficient, Human Development Index (HDI), municipal HDI, HDI-income, or HDI-education, and their association with periodontal health, have not been thoroughly explored. Recently, a reverse gradient in PD according to the Human Development Index has been revealed as a consequence of inequalities in the distribution of the burden of oral conditions [28,29,30,31]. The HDI was introduced in the 1990s by the United Nations Development Program as an instrument to evaluate the development level of regions and to help countries identify living conditions and health inequalities [32,33,34].

The Social Development Index (SDI) is a measure used in Mexico to assess the level of well-being and social progress in the country. It was created in the early 2000s based on the HDI. It was designed to classify the degree of social development of territorial units, geographic spaces that correspond to the subdivision of the municipal geostatistical areas in Mexico City. The SDI is calculated based on a methodology developed by the National Council for the Evaluation of Social Development Policy (Consejo Nacional de Evaluación de la Política de Desarrollo Social, CONEVAL) (for further details on the SDI, see Methods) [35].

Many efforts have been made to evaluate periodontal conditions accurately through self-reported symptoms [18,36,37,38,39,40]. Self-report questionnaires have acceptable validity and can be used for epidemiological studies regarding the surveillance of PD because they are less demanding in terms of cost and time and can be supplemental in the evaluation of the association between PD and other conditions [41,42,43,44]. The choice to apply a self-report questionnaire to communicate the periodontal conditions of a population makes sense within public health settings when the conditions for a clinical evaluation, such as periodontal probing, is not possible, as happened in the COVID-19 pandemic emergency [45,46,47,48,49].

The objectives of this study are then: (1) to detect the presence of some relevant self-reported oral health characteristics, (2) to explore how they are distributed among the different strata of the SDI, and then (3) to determine the association both of the SDI and the SDI’s indicators with some periodontal conditions in a group of adult volunteers from Mexico City.

## 2. Materials and Methods

### 2.1. Study Population and Data Collection

This study was designed as an epidemiological, observational, and self-reported survey of oral health conditions (see Figure 1). The study population consisted of mentally competent volunteer adults over 20 years of age who participated in a cohort from Mexico City. This prospective longitudinal study aimed to explore periodontal clinical conditions in relation to cardiovascular risk factors previously identified at the beginning of the study by our group of researchers [50,51,52].

Volunteers who wished to participate were contacted by email to inform them of the study’s purpose and obtain their consent to complete a standardized questionnaire on general health. The questionnaire included questions about the presence of concomitant systemic diseases, the daily intake of medication prescribed by a treating physician, a self-report of their periodontal condition, their self-perception of gingivitis, dental care, and oral hygiene habits, and diet changes due to chewing problems, among others [53]. The questions related to oral and periodontal health had already been included in validated questionnaires in different populations before [54,55,56,57,58,59,60,61,62,63,64,65,66].

The questionnaire was administered between May and July 2021, well after the second wave (i.e., the largest infection peak) in Mexico [67]. The invitation to participate was sent via email, with up to three reminders sent to those who had not responded. The first email was sent on 28 May 2021, followed by emails on 14 June 2021, and 12 July 2021. The email explained the study’s objective, assured anonymity and confidentiality of responses, requested consent to participate, and ended with a thank-you message for their participation. The questionnaire was designed to take approximately 10 min to complete. The oral health questionnaire consisted of 21 items grouped into four sections: periodontal conditions, dental treatment, oral hygiene, and ability to chew (see Table A1 and Table A2). Questionnaires with incomplete information or duplicates sent by the same email account were discarded. In the latter case, the first questionnaire received was chosen as long as it was complete, or subsequent ones were chosen with the same criteria.

Some items regarding demographic information such as sex, age, and educational level were also recorded. Educational level was classified into five categories: elementary school, middle school, high school, university studies and postgraduate school. The COVID-19 diagnosis was recorded as positive if the person answered affirmatively to the following question: Have you tested positive for COVID-19?

Furthermore, health and lifestyle history related to metabolic syndrome (MetS), as defined by the National Cholesterol Education Program Adult Treatment Panel III (NCEP-ATPIII) criteria; physical activity, which was classified by the International Physical Activity Questionnaire by metabolic equivalents minutes/week in the following categories: low, moderate, and high; sleep quality, defined by the Medical Outcomes Study Sleep scale of 12 items; and current tobacco or cigarette use were collected from previous interviews and clinical sessions carried out previously as part of the follow-up of the studied cohort [51,52].

### 2.2. Institutional Review Board Statement

This study was approved by the Research and Ethics Committee of the School of Dentistry, Universidad Nacional Autónoma de México (CIE/0308/05/2019).

### 2.3. Social Development Index

The SDI is made up of several main dimensions related to education, health, and housing. Each dimension is evaluated using a set of specific indicators, with each indicator weighted differently based on its relative importance in the overall measurement of social development. The resulting scores are combined to obtain a score for each dimension (see Table 1). The value of the SDI allows the territorial units to be ordered according to the levels of development they have reached, which are classified as very low, low, medium, and high (see Table 2) [68].

The SDI indicators are briefly described below [35]:Quality and available space in the home (QASH): The quality of housing is measured by the type of flooring, and the amount of living space is indicated by the number of people per bedroom, with two being the standard.Educational access (EduAcc): This indicator measures the proportion of people aged 18–59 who have completed secondary school or have received 13 years of schooling, which is considered a minimum standard for well-being.Access to social security and/or medical services (ASSMS): This indicator measures the coverage of any of the Mexican health systems.Durable goods (DGd): This indicator measures possession of material goods whose value is equal to or greater than 17.81 USD, or possession of at least three items such as a television, gas stove, computer, refrigerator, or washing machine.Sanitary adequacy (SAd): This indicator measures the availability of a water supply, toilet facilities, and access to a drainage system.Electricity access (EAcc): This indicator measures whether or not there is adequate access to electricity.

Explicit SDI calculation proceeds in five stages, described as follows:

Indicator selection: For each dimension of the IDS, several indicators are selected that reflect the level of development in that area. For example, in the education dimension, indicators such as the percentage of people who have completed secondary education, the literacy rate, and enrollment in higher education can be included.

Establishment of criteria: For each indicator, criteria are established that define the different levels of development. For the indicator of enrollment in higher education, criteria can be established based on the percentage of the population that has completed higher education as very low, low, medium and high.

Assignment of scores: A score is assigned to each indicator based on the level of development achieved. For example, if the percentage of the population that has completed higher education is 20%, a low or moderate score can be assigned, while a percentage of 80% or more could receive a high score.

Calculation of scores by dimension: Once scores have been assigned to each indicator, an average score is calculated for each dimension. For example, if the education dimension has five indicators, the scores for each one are combined by weighted averaging them to obtain the final score for the education dimension.

Combination of scores: Finally, the scores from the different dimensions are combined to obtain the final score of the SDI. Each dimension is weighted differently depending on its relative importance in the overall measurement of social development.

SDI is calculated by means of a Z-score standardized measure for each indicator selected in the different dimensions. A given Z-score indicates the relative position of a value compared to the population.

The Z-score is calculated for each indicator using the following formula:(1)$$Z-\mathrm{score}\;=\;\frac{\mathrm{Indicator}\;\mathrm{value}-\mathrm{State}\;\mathrm{mean}}{\mathrm{State}\;\mathrm{standard}\;\mathrm{deviation}}$$
where Indicator value represents the observed value for the indicator in question in a given region and State mean and State standard deviation represent the mean and standard deviation of the indicator at the level of the given region, respectively.

Finally, the Z-score is standardized so that the range of values is between 0 and 1, using the following formula:(2)$$Z^{std}\;=\;\frac{Z+2}{4}$$

This standardized Z-score is used to evaluate the level of social development in each state and throughout the country. The standardized Z-score allows for the comparison of values for different indicators and dimensions, which helps identify the areas that require the most intervention to improve the level of social development in Mexico.

Z-scores are used to measure relative poverty in each dimension, meaning that they compare the level of social development of each state to the national average in that dimension. For each dimension, a poverty threshold is established using the Z-score corresponding to the 40th percentile of the national distribution. Any state with a Z-score below this threshold is considered to have a low level of social development in that dimension and therefore a higher incidence of poverty in that area. For example, if the poverty threshold in the education dimension is set at a Z-score of −1.28 (the 40th percentile of the national distribution), any state with a Z-score below −1.28 would be considered to have a higher incidence of poverty in education. Similarly, poverty thresholds are established in the other dimensions using the same method.

The SDI has contributed to generating information related to life circumstances and health conditions, such as cardiovascular health [50]. However, few studies have investigated the association between periodontal condition and the indicators that make up social development [26,69], and to the best of our knowledge, no studies have investigated the association between periodontal condition and the social development index specifically in Mexico.

**Table 1 healthcare-11-01548-t001:** Indicators and weights of the Social Development Index.

Indicators	Abbreviation	Weigh
Quality and available	QASH	0.338
space in the home		
Educational access	EduAcc	0.244
Access to social security	ASSMS	0.291
and/or medical services		
Durable goods	DGd	0.060
Sanitary adequacy	SAd	0.038
Electricity access	EAcc	0.029

**Table 2 healthcare-11-01548-t002:** Levels and range of values of the Social Development Index.

Value	Level
<0.7000	Very low
0.7001–0.8000	Low
0.8001–0.9000	Medium
>0.9001	High

### 2.4. Statistical Analysis

The distribution of numerical data was assessed using the Shapiro–Francia tests; the assumption of normality of the data was met with (*W’* > 0.05). Numerical data are presented as median and interquartile range (IQR, Q1–Q3), and qualitative variables are presented as absolute values and percentages (n, %). Demographic characteristics (sex, age, and education level), variables related to health in general (COVID-19, MetS, smoking, physical activity, and sleep quality) and variables related to oral health (periodontal conditions, dental treatment, oral hygiene, and ability to chew) were calculated to determine the differences between the levels of social development index.

After assessing for clinical relevance and practicality, the selected dependent variables, such as have gum disease, loose tooth, loose tooth later, bleeding gums, bad breath and bone loss were incorporated into multivariate logistic regression models to estimate the association between periodontal conditions and some associated variables including the global SDI and its indicators (see Table A3). Odds ratios (OR) and 95% confidence intervals (2.5% C.I.–97.5% C.I.) were calculated to estimate the strength of association. All tests use a confidence level α=0.05.

The variance inflation factor (VIF) was determined for each regression model to assess for multicollinearity. All retained variables had VIF values less than 10. Stepwise regression was performed using maximum likelihood calculations to select the best model. The Wilks test, based on the Neyman–Pearson lemma that optimizes considering both types of error, was used as the maximum likelihood criterion in the likelihood ratio test. The balance between sensitivity and specificity was optimized using ROC curves and calculation of the area under the ROC curve (AUROC). Analysis of residuals and homoscedasticity was also performed on all regression models. Analyses were performed using R version 4.0.2 [70]. Log-likelihood ratios were calculated to assess goodness of fit for the models. The Akaike Information Criterion (AIC) was used to estimate the model’s likelihood of predicting future values. Lower AIC scores indicate a better fit for the model.

## 3. Results

### 3.1. General Characteristics

This study includes information from 1294 healthy adults with a median age of 43 years (IQR 34-50). Women accounted for 65.15% of the participants, and 19.86% self-reported a COVID-19 diagnosis. University was the most common level of education (48.92%). Health-related conditions such as smoking (19.01%), low physical activity (18.47%), and poor sleep quality (45.60%) were also reported. Age and COVID-19 diagnosis were found to be statistically significant when comparing between levels of social development ($p_{value}<0.050$ and $p_{value}<0.001$, respectively); see Table 3.

### 3.2. Self-Reported Oral Health

#### 3.2.1. Self-Reported Periodontal Conditions

Around 50% of the studied population reported having bleeding gums when brushing their teeth (see Table 1 for full feature names and explanations); when comparing between SDI levels, there was a statistically significant difference in affirmative self-reported answers ($p_{value}<0.050$). Around 40% reported having lost a tooth without a previous injury; this was an issue that showed similar distributions in self-report between SDI levels ($p_{value}\;=\;0.864$) but with an apparent decrease in the percentage at a higher level of development (see Table 4).

Approximately 32% of the participants noticed that a tooth did not seem right in the past three months ($p_{value}<0.001$), 25% thought they had gum disease ($p_{value}<0.050$), around the same proportion (24.80%) reported having bad breath ($p_{value}<0.001$), 20% of the participants thought their gums had been injured or infected in the last year ($p_{value}<0.050$), and less than 15% of the participants reported having a dental abscess in the same time period ($p_{value}<0.001$). Regarding the question, has a dentist ever told you that you have bone loss around your teeth? a positive incremental presence of responses related to the level of SDI was found, as well as statistically significant differences between the groups compared ($p_{value}<0.050$; see Table 4).

#### 3.2.2. Self-Reported Dental Treatment, Oral Hygiene, and Ability to Chew

Approximately 52% of the participants have received dental care in the last 12 months. Although no statistically significant differences were found in the distribution of dental care between SDI levels, participants at the highest levels reported receiving dental care more often than those at lower levels ($p_{value}\;=\;0.333$). Around 22% reported having gum treatment in the last year, and the differences were statistically significant when compared between SDI levels ($p_{value}<0.050$); see Table 4).

Regarding self-reported oral hygiene, more than 97% of participants responded positively to usually tooth brushing ($p_{value}\;=\;0.881$), and the majority reported brushing their teeth ≥ 2 times per day ($p_{value}\;=\;0.061$). A total of 80.91% of participants brush their teeth before going to bed ($p_{value}<0.050$), with a positive increase at a higher level of development reported. Only 16.77% used floss after teeth brushing, with the same incremental pattern related to the SDI level. There were no statistically significant differences between the compared groups in terms of how often they reported brushing their teeth per day or using mouthwash per week. However, daily use of dental floss was found to be statistically significant, and this behaviour increased with higher levels of SDI (see Table 4).

In terms of self-reported ability to chew, most participants reported being either satisfied or very satisfied (54.10% and 35.24%, respectively). Both levels showed statistically significant differences in relation to SDI levels ($p_{value}<0.001$) (see Table 4).

### 3.3. Associations of Self-Reported Periodontal Conditions

After estimating various multivariate logistic regression models, we selected the two presented below: (1) associations with bone loss and the overall SDI, and (2) associations with bone loss and the indicators related to the SDI, including either one to avoid collinearity between variables. The full maximum likelihood, adjusted odds ratios (95% CI), and *p*-values of the final models are presented in Table 5 and Table 6, respectively.

Bone loss was the dependent variable in the multivariate regression model with the global SDI included (see Table 5) mainly associated with Times Floss use ($p_{value}=3.3\times 10^{-4}$) (see Table A1 for full feature names and explanation). The global SDI presented a significant association with it ($p_{value}=0.025$), and participants who scored the highest overall SDI value had a 7.27 times higher chance of having ever been told by a dentist that they were losing bone around their teeth. Individuals who reported low physical activity ($p_{value}=0.013$) and moderate physical activity ($p_{value}=0.005$) had a 50% and 42%, respectively, lower probability of having been diagnosed with bone loss compared to those who do not have these features. As expected, participants who were very unsatisfied with their chewing ability ($p_{value}=4.75\times 10^{-4}$) had up to 5.87 times the probability of being diagnosed by a dentist as losing bone around their teeth. The AUROC was 0.6898706.

On the other hand, bone loss as a dependent variable in the multivariate regression model with the SDI indicators included (see Table 6) has also been associated with the times floss use ($p_{value}=3.14\times 10^{-4}$). However, the SDI component most strongly associated with bone loss was the QASH: Quality and available space in the home ($p_{value}=0.028$), and those participants who scored the highest QASH value had a 3.66 times higher chance that they had ever been told by a dentist that they were losing bone around their teeth. Subjects with self-reported low physical activity ($p_{value}=0.014$) and moderate physical activity ($p_{value}=0.005$) were also found to have a 50% and 42%, respectively, lower probability of having bone loss diagnosed compared to those without diagnosed bone loss. Additionally, those participants who were very unsatisfied with their chewing ability ($p_{value}=4.88\times 10^{-4}$) had up to a 5.85 times higher probability that they had ever been told by a dentist that they were losing bone around their teeth. The AUROC was 0.6898994. The other estimated multivariate models did not show an association between the main periodontal variables (such as loose tooth, loose tooth later, bleeding gums, bad breath, and bone loss) and the global SDI or with its indicators.

## 4. Discussion

Few studies have previously reported on the periodontal status of the inhabitants of Mexico City, and most of these studies have focused on the older adult population (70 years and older) [71,72,73]. To our knowledge, no study has investigated self-reported periodontal conditions related to the level of social development in a Mexican population. In this study, we present the results of a self-report questionnaire that was administered online to young adult volunteers in Mexico City during the COVID-19 pandemic, and we classify the participants by level of social development. The aim of the study is to investigate self-reported periodontal conditions in this population.

### 4.1. General Findings

In brief, we found that bone loss was significantly associated with SDI, both global SDI and with the QASH SDI indicator. It was also significantly associated with physical activity, time floss used, and satisfaction level (for more information on the features and how they were surveyed, please refer to Appendix A and see Table 5 and Table 6).

Our results indicate that people in the very low SDI level present much higher rates of self-reported perception of at least one tooth that does not seem right in the last three months than those in the higher SDI levels. However, this group reported receiving dental care less frequently than those in the upper level. This finding might suggest that some residents of Mexico City in a low SDI stratum are aware of having a periodontal problem but overlook it because they are limited in seeking dental care due to the possible lack of social protection for oral health. This scenario would explain the higher frequency of reported advanced periodontal disease clinical features, such as loss of teeth, loose teeth later, and bleeding gums. Survey respondents with lower SDI status tend to lag in carrying out good oral hygiene practices such as flossing after brushing teeth or brushing teeth before bed.

Consistent with our findings, some studies have mentioned that self-reported periodontal clinical features (bleeding gums, tooth mobility, and tooth loss) are associated with lower social stratum factors [18,74]. It has also been reported that less familiarity with proper tooth brushing techniques and the use of oral care products is related to a lower level of socioeconomic status [59,75]. Bleeding gums is an early sign of gingival inflammation. It is the most straightforward self-recognition symptom by which subjects can auto-report and/or self-diagnose a present periodontal disease [59,62]. In turn, self-awareness of periodontal health status may influence oral-health-care-seeking behaviour, though this can be challenged by limited access to dental care, which is a known marker of health inequality, particularly in the context of the recent COVID-19 pandemic [76,77,78,79].

In contrast, people in medium or higher SDI groups appear to experience early identification of pathological periodontal conditions, as possibly explained by the affirmative answers to the following questions: Do you think you may have gum disease? or During the last year, have your gums been injured or infected? This was also indicated by the affirmative answer to Have you ever had a gum treatment such as scraping or root planing, which is sometimes referred to as deep cleaning? Those subjects who received a warning of loss of bone were perhaps the ones who reported better dental care and better oral hygiene practices due to their high SDI levels.

An outstanding result of this study is the association of global SDI and one of its indicators, the quality and available space in the home, which is related to one of the leading clinical features of periodontal disease (bone loss). This positive association may be due to the fact that people with a higher SDI have better life conditions, which allows them to establish good oral hygiene practices. Likewise, the positive association of QASH with bone loss may be related to the fact that those people with favorable SDI may identify clinical features of periodontal disease in a timely manner and have access to specialized periodontal care. QASH is a quite comprehensive and complex variable and can encompass a variety of vulnerable situations that can modify conditions necessary to keep good oral health [80,81,82,83,84].

Adverse housing conditions are not a new problem for social development in the Latin American region [85]. Although significant progress has been made in recent decades, there are still gaps in the management system and the quality of public services [85,86,87]. In Mexico, housing factors are some of the main social determinants of health [88]; however, the implementation of policies to mitigate these factors has been slow [89]. The harsh conditions of social development are further affected by challenging barriers to accessing public healthcare and social security [90,91,92]. Regrettably, this situation is prevalent worldwide [93], and scant literature has considered the joint effects of social development on periodontal disease [94].

Few studies have, however, evaluated inequalities in self-reported use of oral hygiene products between different income or socioeconomic levels [95,96,97], and to the best of our knowledge, none have explored the association with the parameters of social development. On the other hand, several studies have analyzed the effects of the routine use of dental floss on plaque or any of the clinical parameters of periodontal disease without establishing any potential benefit [98]. A significant association between bone loss and oral hygiene practices, such as that seen in times floss use, was observed in our regression models (Table 5 and Table 6). The positive association found here may originate from bias due to the relatively higher SDI of the participants.

Regarding the association we found between bone loss and physical activity, some studies have concluded that physical activity can improve glycemic control, endothelial function, peripheral blood flow, and the systemic inflammatory profile. These mechanisms, in turn, can attenuate some clinical features of periodontal disease, such as alveolar bone loss [99,100,101,102,103,104,105,106]. However, we did not obtain periodontal clinical data, such as probing pocket depth, clinical gingival recession, or attachment loss (see the Limitations of the present study subsection below).

### 4.2. Implications for Health Policy

Periodontal diseases, similar to many oral diseases, are largely preventable, yet they persist with a high prevalence, reflecting some neglected social inequalities as well as the poorest health care access [1]. Some periodontal conditions disproportionately affect socially disadvantaged people and may be associated with other chronic diseases. Our findings reveal that a consistent social gradient exists between SDI levels and the prevalence and severity of periodontal conditions. In this sense, periodontal diseases can be considered a marker of socioeconomic disadvantage and a sign of the ill health conditions of a population linked to unmet needs for social development [107]. The persistence of oral health disparities indicates that population-based policies are urgently needed to address the underlying social, economic, and environmental causes of oral diseases, such as housing and social protection [10]. This points to the need to plan strategies to identify high-risk individuals through screening. Additionally, improving the life-course approach to periodontal health services could help everyone seek treatment in the early stage of the disease [10,108].

It is important to note that PD are associated with multiple ailments [8]. One significant example is diabetes, as oral hygiene practices and the presence of PD have been found to be linked to diabetes and its outcomes [109,110]. This relationship must be taken into account when planning health policies. Oral hygiene and periodontal treatment can help control biofilm bacterial dysbiosis, which can reduce gingival inflammation. However, severe periodontitis patients have a higher prevalence of prediabetes, diabetes, and metabolic disorders, such as dyslipidemia, in Type 2 diabetes patients [110]. Recent studies suggest that innovative treatments, such as irrigation of periodontal pockets with ozonated water in the dental office, rinsing with a certain percentage of ozone, and using specialized toothpaste and supplements for precise interdental hygiene at home, could be useful for diabetic patients in controlling and removing the oral bacterial biofilm. These treatments have been shown to be effective in maintaining glycated hemoglobin values within acceptable limits [109].

Regarding our COVID-19-related findings, it is worth highlighting that statistically significant differences were observed between individuals residing in areas with very low and low SDI compared to those living in areas with a median SDI, particularly in terms of age (40 and 43 years, respectively), prior COVID-19 diagnosis, and education level (middle and high school). These differences may be attributed to the fact that individuals with lower socioeconomic status have had less access to healthcare, as previously reported in marginalized regions of Mexico City [111].

As already mentioned, several studies have considered the hypothesis of a possible association, even a causal role, between periodontitis and COVID-19 [112,113,114]. Their inflammatory and infectious natures have been reported to be among the mechanisms linking both diseases [115,116]. Therefore, external factors such as discontinuing regular dental visits and neglecting oral health could increase symptoms of periodontitis and COVID-19 [117]. Some of these factors are mediated by socioeconomic restrictions or income contraction, leading to a deterioration of physical and oral health during the pandemic [118,119]. The impacts of COVID-19 on oral health at the practice level are extensively described, as documented by Dickson-Swift and collaborators [120]. However, gaps remain in understanding the impact on individual oral health, particularly on periodontal conditions. Therefore, our interest in including the positivity of COVID-19 in the study was to evaluate the possible contribution of this variable to the association between self-reported periodontal symptoms and the level of social development in the CDMX population, thus paving the way for future investigations on this relationship.

### 4.3. Limitations of the Present Study

The present study has several limitations. For instance, due to the conditions present during the COVID-19 pandemic, the questions were not validated with a clinical examination, such as radiographic bone level or periodontal probing depth. Only internal consistency assessments were performed, and as such, the results of this study cannot be extrapolated to all young adults in Mexico City.

Additionally, the possible recall bias of the participants, or a potential social desirability bias in reporting the frequency of tooth brushing, needs to be considered. It is believed that respondents tend to over-report behaviors they consider more desirable for the purpose of the study [4]. This is particularly relevant since this study reached a highly educated segment of the Mexican population (for instance, 48.92% of our subjects reported having university studies). Furthermore, since we conducted an online survey, the subjects had, by definition, internet access, which can be considered an implicit inclusion criterion.

On a positive note, we would like to highlight the prompt response rate of the participants, as well as the significance of this study as the first to be conducted on a young urban population in Mexico City. Despite the inability to perform a clinical examination due to the COVID-19 pandemic, this study has provided valuable insights into the prevalence of and treatment needs regarding periodontal disease in this population.

### 4.4. Perspectives

The present work is part of a broader research framework that includes studying the associations of periodontal disease with other diseases, such as hypertension [53] and diabetes, as well as the association of PD with social determinants of health and risk factors and its relationships with chronic inflammation and immunological factors (a work in progress). Regarding this particular branch project, our following steps would be to validate the self-report periodontal symptoms via clinical exploration and measurement sessions.

## 5. Conclusions

Periodontal diseases are a set of complex oral health conditions. Such conditions arise from the interplay of genetic, microbiological and other environmental features. Particularly relevant to efforts in oral public health is the relationship between social determinants of health and periodontitis. The Social Development Index is a systematic quantitative tool developed in Mexico to analyze and evaluate social disparities with a view toward policy development.

The study confirms the existence of a relationship between self-reported periodontal conditions, such as bone loss and global SDI, as well as bone loss and quality and available space in the home, in the particular COVID-19 pandemic context. Those individuals who ranked lower in social development status reported having COVID-19 and more advanced clinical features of periodontal disease, such as bleeding gums and prevalence of tooth mobility, while those who ranked higher in social development more accurately reported gingival infection as a distinct sign of periodontal disease. In addition to demographic or socioeconomic measures, a combination of social development dimensions and self-reported oral health characteristics could have promising potential in assessing periodontal health status for public health program planning, especially when clinical assessment is unattainable.

Analyzing the landscape of social and environmental risk factors for preventable diseases, such as periodontitis, paves the way to the implementation of goal-oriented policies. In this regard, online questionnaires have shown to be a convenient way to survey large fractions of the population with a view toward targeted interventions. Validating these questionnaires in adequately large cohorts is necessary to further establish the scope and limitations of these studies.

## Figures and Tables

**Figure 1 healthcare-11-01548-f001:**
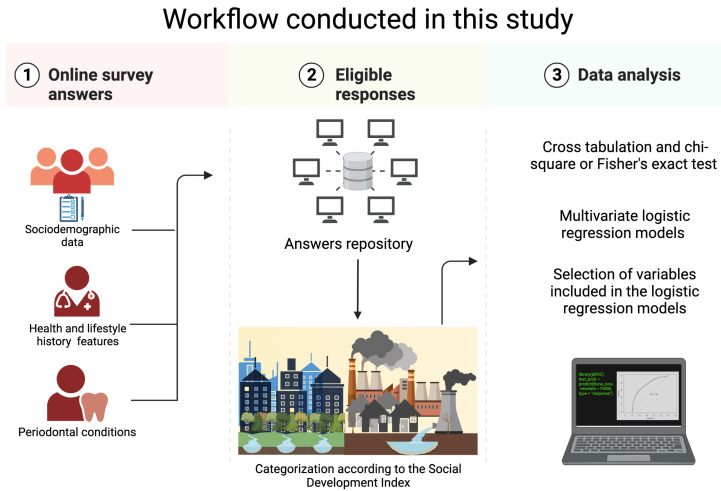
The figure presents the workflow from data collection, processing, and analysis of the data carried out in this study. Created with BioRender.com.

**Table 3 healthcare-11-01548-t003:** Distribution of sociodemographic, health, habit, and lifestyle characteristics according to the Social Development Index levels.

Sociomedical Features	Very Low (n = 134)	Low (n = 462)	Medium (n = 305)	High (n = 393)	*p* Value
Female *	61.94	64.72	65.25	66.67	0.790
Male *	38.06	35.28	34.75	33.33	
Age, years **	40 (32–48)	43 (33–50)	43 (35–50)	45 (36–51)	<0.05
Elementary school *	0.74	0.22	0.66	0.25	0.545
Middle school *	11.94	5.19	4.92	2.54	<0.001
High school *	40.30	36.80	29.18	21.12	<0.001
University studies *	38.81	47.62	50.82	52.42	<0.001
Postgraduate *	8.20	10.17	14.42	23.66	0.031
COVID-19 *	21.64	24.89	19.02	13.99	<0.001
Smoking *	18.66	22.29	16.72	17.05	0.852
MetS diagnostic *	6.72	4.33	3.61	4.58	0.543
Physical activity (Low) *	16.42	19.70	19.34	17.05	0.677
Physical activity (Moderate) *	50	46.10	44.92	48.85	0.636
Physical activity (High) *	33.58	34.20	35.74	34.10	0.959
Sleep quality (Bad) *	38.81	46.54	44.26	47.84	0.299

* Affirmative answer, value expressed as a percentage; ** values expressed in median, IQR 25–75.

**Table 4 healthcare-11-01548-t004:** Distribution of periodontal characteristics according to the Social Development Index levels.

Oral Health Features	Very Low (n = 134)	Low (n = 462)	Medium (n = 305)	High (n = 393)	*p* Value
Periodontal conditions					
Tooth does not look right *	38.06	31.60	33.44	30.28	<0.001
Loose tooth *	11.19	9.09	10.16	10.18	0.885
Loose tooth later *	8.21	5.63	6.56	6.87	0.725
Lost tooth *	44.78	41.77	40.66	40.97	0.864
Gum disease *	19.40	25.54	31.80	25.45	<0.05
Bleeding gums *	48.51	52.38	50.82	47.84	<0.001
Gum infected *	16.42	20.56	19.67	22.14	<0.05
Bad breath *	23.13	25.97	27.54	21.88	<0.001
Abscessed teeth *	14.93	15.80	12.13	12.47	<0.001
Bone loss *	8.96	9.09	13.44	14.50	<0.05
Dental treatment					
Dental care *	45.52	52.38	50.16	54.20	0.333
Implant placed *	14.18	11.47	15.08	12.98	0.517
Gum treatment *	17.16	22.73	25.25	21.88	<0.05
Oral hygiene					
Brush your teeth *	98.51	97.62	97.38	97.96	0.881
Times brush teeth **	2 (1)	2 (1)	2 (1)	2 (1)	0.061
Brush before bed *	76.12	78.14	80.33	86.26	<0.05
Floss use after *	6.72	16.23	20.33	20.61	0.259
Times floss use **	0 (0–2)	1 (0–3)	1 (0–3)	1 (0–4)	<0.001
Times mouthwash use **	0 (0–3)	0 (0–3)	0 (0–3)	0 (0–3)	0.240
Ability to chew					
(Satisfaction level)					
Very unsatisfied *	2.24	1.30	1.97	0.76	0.160
Unsatisfied *	8.96	9.96	9.51	8.40	0.032
Satisfied *	55.97	55.19	47.87	57	<0.001
Very Satisfied *	32.84	33.55	40.66	33.84	<0.001

* Affirmative answer, value expressed as a percentage; ** values expressed in median, IQR 25–75.

**Table 5 healthcare-11-01548-t005:** Logistic regression model considering *Bone loss* as the outcome variable and *global SDI* as one of the independent variables.

Variables	Estimate	OR (95% C.I.)	*p*-Value
Intercept	−3.443	0.032 (−4.955, −1.982)	5.47 $\times 10^{-6}$
SDI global	1.985	7.276 (0.258, 3.744)	0.025
Physical activity (Low) †	−0.683	0.505 (−1.248, −0.159)	0.013
Physical activity (Moderate) †	−0.535	0.586 (−0.913, −0.159)	0.005
Times floss use	0.076	1.079 (0.034, 0.118)	3.3 $\times 10^{-4}$
Satisfaction level (Unsatisfied) ‡	0.736	2.088 (0.231, 1.219)	0.003
Satisfaction level (Very satisfied) ‡	−0.734	0.480 (−1.187, −0.306)	0.001
Satisfaction level (Very Unsatisfied) ‡	1.770	5.869 (0.742, 2.759)	4.75 $\times 10^{-4}$

SDI: Social Development Index. †: Physical activity (high) was the reference category. ‡: Satisfied with ability to chew was the reference category.

**Table 6 healthcare-11-01548-t006:** Logistic regression model considering *Bone loss* as the outcome variable and *SDI indicators* as independent variables.

Variables	Estimate	OR (95% C.I.)	*p*-Value
Intercept	−2.750	0.064 (−3.676, −1.851)	3.34 $\times 10^{-9}$
QASH	1.299	3.666 (0.145, 2.460)	0.028
Physical activity (low) †	−0.681	0.506 (−1.246, −0.157)	0.014
Physical activity (moderate) †	−0.535	0.586 (−0.914, −0.160)	0.005
Times floss use	0.077	1.080 (0.034, 0.118)	3.14 $\times 10^{-4}$
Satisfaction level (unsatisfied) ‡	0.737	2.090 (0.232, 1.220)	0.003
Satisfaction level (very satisfied) ‡	−0.732	0.480 (−1.186, −0.304)	0.001
Satisfaction level (very unsatisfied) ‡	1.766	5.847 (0.739, 2.754)	4.88 $\times 10^{-4}$

QASH: Quality and available space in the home. †: Physical activity (high) was the reference category. ‡: Satisfied with ability to chew was the reference category.

## Data Availability

All data and code used in this study can be found at: https://github.com/CSB-IG/SDI_PD (accessed on 15 October 2022).

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
