# Peer review of "Relationship between the Social Development Index and Self-Reported Periodontal Conditions"

_healthcare, 2023, doi:10.3390/healthcare11111548_

Round 1
Reviewer 1 Report
Thank you so much for allowing me to review this article. There are some comments I would like you to address.
Introduction:
I’m not sure if this statement is correct “PD are the most common multifactorial oral pathology in 20 adults, characterized by the disruption of biofilm ecology [5]”. Untreated dental decay is the most prevalent non transmissible disease worldwide.
In general, introduction is a bit too long. For example, the first paragraph of the introduction contains too many ideas, making it difficult to understand. Please, I would suggest that you tidy up and shorten the introduction better by removing irrelevant information. And also suggest that you finish this section by describing the objectives of the study in a simpler way.
The information on the method used in the data collection should be in the material and methods section, so I suggest you move the information to that section.
Materials and Methods
A section describing the study population would be appreciated under the heading of study settings. some of this information is included in the introduction. In my opinion, this information should be included in this section.
Dear authors, are you sure that figure 1 is necessary?
The information in the third and fourth paragraph of this section is not necessary and makes reading difficult. Please delete or shorten it. I also suggest that you move table 1 and 2 to supplementary material because it does not provide relevant information about the study.
It is not clear how the questionnaire was distributed to the volunteers who participated in the study. Please provide information on this issue in this section.
Results
In this section there is information that should be in the previous section such as lines 226-232. As in the previous section, table 5 does not provide relevant information from the study. I suggest you remove it from the article.
This section is too long and difficult to read, mainly due to unnecessary information such as the variable type (example lines 233-36). Please express p-values correctly, it is not necessary to put so many decimals in the text. the same for ORs, they should be expressed in the text and in the tables as (OR x 95%CI: ). I suggest you look for information on how to express them correctly.
Discussion
I suggest that you start this section by summarising the most relevant results of your research.
This section is also too long and difficult to read, I suggest you summarise and simplify it.
Please check certain statements that are incorrect due to the study design like line 344-45.
Author Response
The authors want to thank Reviewer 1 for their review and professional critique of our work. In what follows we are presenting a point-by-point response (in bold type) to your comments, concerns and suggestions. A "manuscript with tracked changes" file has been included for your convenience.
(x) English very difficult to understand/incomprehensible
Thank you so much for allowing me to review this article. There are some comments I would like you to address.
Introduction:
I’m not sure if this statement is correct “PD are the most common multifactorial oral pathology in adults, characterized by the disruption of biofilm ecology [5]”. Untreated dental decay is the most prevalent non transmissible disease worldwide.
In general, introduction is a bit too long. For example, the first paragraph of the introduction contains too many ideas, making it difficult to understand. Please, I would suggest that you tidy up and shorten the introduction better by removing irrelevant information. And also suggest that you finish this section by describing the objectives of the study in a simpler way.
As reviewer 1 mentions, untreated dental caries is the most prevalent noncommunicable disease worldwide. Our statement now specifies that periodontitis is a major public health problem due to its high prevalence, and that it has become the most common non-communicable chronic inflammatory disease in humans.
We have reworked the introduction section to provide a clearer take of the full scope and the main objectives of our study. Additionally we added information on the Social Development Index to the introduction as requested by reviewer 2.
Now the objectives of our study read as follows:
The objectives of this study were: 1) to detect the presence of some relevant self-reported oral health characteristics, 2) to explore how they are distributed among the different strata of the SDI and then 3) to determine the association both of the SDI and the SDI's indicators with some periodontal conditions in a group of adults from Mexico City.
The information on the method used in the data collection should be in the material and methods section, so I suggest you move the information to that section.
The information related to the data collection was moved to the material and methods section as the reviewer suggested.
Materials and Methods
A section describing the study population would be appreciated under the heading of study settings. Some of this information is included in the introduction. In my opinion, this information should be included in this section.
The information related to the study population has been incorporated into the section suggested by the reviewer, thank you.
Dear authors, are you sure that figure 1 is necessary?
We would like to retain Figure 1. It was originally devised as the journal's required graphical abstract. Although it is quite simple, we think that it serves as an overview.
The information in the third and fourth paragraph of this section is not necessary and makes reading difficult. Please delete or shorten it. I also suggest that you move table 1 and 2 to supplementary material because it does not provide relevant information about the study.
The information pointed out by reviewer 1 has been shortened, thanks for the suggestion. Regarding Tables 1 and 2, as well as Table 5, we agree with the reviewer that they were placed in such a way that the reading flow of the manuscript was somehow interrupted. However, we consider that having the questionnaires in which the study is based in the manuscript, instead of placing them in a supplementary document is good for consistency. Hence, we decided to add them in the form of an appendix (now tables A7, A8 and A9) so that they do not interrupt the reading and at the same time are accessible within the manuscript.
It is not clear how the questionnaire was distributed to the volunteers who participated in the study. Please provide information on this issue in this section.
Thanks for the observation. The paragraph has been rewritten to clarify how the questionnaire was distributed to the volunteers who participated in the study.
Results
In this section there is information that should be in the previous section such as lines 226-232. As in the previous section, table 5 does not provide relevant information from the study. I suggest you remove it from the article.
This information was moved to the material and methods section. Table 5 was placed in an appendix at the end of the manuscript (now Table A9).
This section is too long and difficult to read, mainly due to unnecessary information such as the variable type (example lines 233-36). Please express p-values correctly, it is not necessary to put so many decimals in the text. the same for ORs, they should be expressed in the text and in the tables as (OR x 95%CI: ). I suggest you look for information on how to express them correctly.
The p-values, as well as the annotation of the OR and 95% CI have been modified both in the text and in the tables by following the reviewer’s suggestions.
Discussion
I suggest that you start this section by summarising the most relevant results of your research.
To clarify this point, we add in the first part of the discussion the main results of the study as well as the changes in the objectives suggested by the reviewer.
This section is also too long and difficult to read, I suggest you summarise and simplify it.
A summary has been added at the beginning of the discussion by following the Reviewer’s advice.
Please check certain statements that are incorrect due to the study design like line 344-45.
Thank you, we reworded this text to avoid methodological inconsistencies, and also corrected some typing and grammatical mistakes.

Reviewer 2 Report
Dear Authors,
thank you for submitting your paper.
It's very interesting because it addresses a very important social topic.
Please remember to cite the relationship between periodontal diseases and diabetes and the importance of oral hygiene.
Butera A, Lovati E, Rizzotto S, Segù M, Scribante A, Lanteri V, Chiesa A, Granata M, Rodriguez Y Baena R. Professional and home-management in non-surgical periodontal therapy to evaluate the percentage of glycated hemoglobin in type 1 diabetes patients. International Journal of Clinical Dentistry, 2021, 14(1), pp. 41–53
Did you ask an Ethic Committee evaluation?
Your study is very important to plan future interventions of prevention in the general population.
Author Response
The authors want to thank Reviewer 2 for their review and professional critique of our work. In what follows we present a response to your comments (in bold type), concerns and suggestions. A "manuscript with tracked changes" file has been included for your convenience.
(x) English language and style are fine/minor spell check required
Dear Authors,
Thank you for submitting your paper. It's very interesting because it addresses a very important social topic.
Please remember to cite the relationship between periodontal diseases and diabetes and the importance of oral hygiene.
Butera A, Lovati E, Rizzotto S, Segù M, Scribante A, Lanteri V, Chiesa A, Granata M, Rodriguez Y Baena R. Professional and home-management in non-surgical periodontal therapy to evaluate the percentage of glycated hemoglobin in type 1 diabetes patients. International Journal of Clinical Dentistry, 2021, 14(1), pp. 41–53
In agreement with the reviewer 2 comment, we added details about the relation between periodontal diseases, diabetes and the importance of oral hygiene in the discussion and incorporated appropriate references including the suggestion of the reviewer.
Did you ask an Ethic Committee evaluation?
R: Yes, the study was approved by the Research and Ethics Committee of the School of Dentistry, Universidad Nacional Autónoma de México (CIE/0308/05/2019). The declaration is in the subsection Institutional Review Board Statement on Materials and Methods.
Your study is very important to plan future interventions of prevention in the general population.
Thank you! We hope our work would be useful for targeted health policy.

Reviewer 3 Report
The authors conducted an epidemiological study on the Mexican adult population based on a questionnaire during the 2021 COVID-19 pandemic. The most important result was the correlation of both self-reported periodontal diseases and diagnosed bone loss with the quality and available space in the home as an indicator of social inequities that negatively influence health. The manuscript has social value, and the Mexican authorities should acknowledge the results of the current research. Anyway, the manuscript is difficult to read, and the latter is a critical obstacle to the understanding and widespread of the social contents. The authors should remember that a paper and its abstract must contain all the key data for understanding them without the need for reading other papers. For example, the abstract is unclear, lacks conclusions, and is impossible to understand without reading the entire manuscript. In addition, the abstract provides unclear information about the study outcome computed in the multivariate logistic regression. This reviewer kindly suggests rewriting the abstract to make it focused on the key results of the research. The introduction section is long but lacks an explanation of the Social Development Index (SDI), which occurs only in a paragraph of the methods section and lacks the reporting of the score ranges for the definition of each social level. Saying “Full details can be found at [52]” is an insufficient explanation because those details regard the MAIN OUTCOME of the study. The addition of a table explaining the entire SDI system can improve the readability of the introduction. The methods section can be improved by adding information such as how the authors selected the volunteers, the period of importance for positivity to COVID-19, and how the authors retrieved the information regarding the metabolic syndrome diagnosis criteria. Another limitation of the study is that the authors reached a part of the Mexican population that was highly educated (48.92% reported university studies) and surely had a connection to the internet, which was an untold inclusion criterion of the current study. Therefore, the authors should acknowledge that those results show the edge of the iceberg of the social inequities in the Mexican population. The conclusion section is supported by the results.
Before publication, the manuscript needs English editing.
In my opinion, the study should be considered for publication after MAJOR REVISIONS.
With warm regards
Author Response
The authors want to thank Reviewer 3 for their professional review and critique of our work. In what follows, we are presenting a point-by-point response (in bold type) to your comments. A "manuscript with tracked changes" file has been included for your convenience.
(x) English language and style are fine/minor spell check required
The authors conducted an epidemiological study on the Mexican adult population based on a questionnaire during the 2021 COVID-19 pandemic. The most important result was the correlation of both self-reported periodontal diseases and diagnosed bone loss with the quality and available space in the home as an indicator of social inequities that negatively influence health. The manuscript has social value, and the Mexican authorities should acknowledge the results of the current research.
Anyway, the manuscript is difficult to read, and the latter is a critical obstacle to the understanding and widespread of the social contents. The authors should remember that a paper and its abstract must contain all the key data for understanding them without the need for reading other papers. For example, the abstract is unclear, lacks conclusions, and is impossible to understand without reading the entire manuscript. In addition, the abstract provides unclear information about the study outcome computed in the multivariate logistic regression. This reviewer kindly suggests rewriting the abstract to make it focused on the key results of the research.
The introduction section is long but lacks an explanation of the Social Development Index (SDI), which occurs only in a paragraph of the methods section and lacks the reporting of the score ranges for the definition of each social level. Saying “Full details can be found at [52]” is an insufficient explanation because those details regard the MAIN OUTCOME of the study. The addition of a table explaining the entire SDI system can improve the readability of the introduction. The methods section can be improved by adding information such as how the authors selected the volunteers, the period of importance for positivity to COVID-19, and how the authors retrieved the information regarding the metabolic syndrome diagnosis criteria. Another limitation of the study is that the authors reached a part of the Mexican population that was highly educated (48.92% reported university studies) and surely had a connection to the internet, which was an untold inclusion criterion of the current study. Therefore, the authors should acknowledge that those results show the edge of the iceberg of the social inequities in the Mexican population. The conclusion section is supported by the results.
As reviewer 3 has suggested, we have now included in the introduction section a further explanation of the Social Development Index. In the materials and in methods section we have reported in detail how the study participants were selected. In the study limitations section, we have incorporated the reviewer's observation related to the bias regarding educational level and Internet connection of the respondents.
Regarding the period of importance for positivity to COVID-19, this study data acquisition was carried out during May - July 2021, in the epidemiological weeks 90 to 105 of the pandemic, well after the second wave (i.e. the largest infection peak) in México. (Benítez-Pérez, H., Herrera, L. A., López-Arellano, O., Revuelta-Herrera, A., Rosales-Tapia, A. R., Suárez-Lastra, M., Hernández-Lemus, E., ... & Ruiz-Gutiérrez, R. (2022). Probability of hospitalisation and death among COVID-19 patients with comorbidity during outbreaks occurring in Mexico City)
Regarding the information of the metabolic syndrome diagnosis, our group recovered this information from interviews and clinical sessions carried out previously as part of the follow-up of the studied cohort (Gutiérrez-Esparza, G. O., Infante Vázquez, O., Vallejo, M., & Hernández-Torruco, J. (2020). Prediction of metabolic syndrome in a Mexican population applying machine learning algorithms. Symmetry, 12(4), 581; Gutiérrez-Esparza, G. O., Ramírez-delReal, T. A., Martínez-García, M., Infante Vázquez, O., Vallejo, M., & Hernández-Torruco, J. (2021). Machine and Deep Learning Applied to Predict Metabolic Syndrome without a Blood Screening. Applied Sciences, 11(10), 4334.)
Before publication, the manuscript needs English editing.
In my opinion, the study should be considered for publication after MAJOR REVISIONS.
We have double checked our manuscripts writing for conceptual clarity. Afterwards our manuscript was sent to a professional English language copy-editor and proofreading service. We have also revised carefully the proofreader-revised version.

Reviewer 4 Report
The article is well structured and the research is conducted in accordance with scientific requirements. I appreciated the idea of the authors, it is an interesting study and I think it can be a starting point for other research in this direction.
I have some questions/recommendations to clarify some of the authors' less-emphasized points.
1. Do you think that the background of the respondents has an influence on the results?
2. I did not find the limits of the study specified
3. What are the criteria for excluding respondents?
4. What is the reason why you introduced COVID-19 as the independent variable? How does it influence the results?
5. The references chapter does not fully respect the recommended style
6. In my opinion, the citation index is much too high and should be revised. (I attach the Semplag report)
7. Are you thinking of continuing your research and if so, in what way?
Author Response
The authors want to thank Reviewer 4 for their review and professional critique of our work. In what follows we are presenting a point-by-point response to your comments, concerns and suggestions (in bold type). A "manuscript with tracked changes" file has been included for your convenience.
(x) I am not qualified to assess the quality of English in this paper
The article is well structured and the research is conducted in accordance with scientific requirements. I appreciated the idea of the authors, it is an interesting study and I think it can be a starting point for other research in this direction.
I have some questions/recommendations to clarify some of the authors' less-emphasized points.
- Do you think that the background of the respondents has an influence on the results?
That's right, the background of the respondents, for example, socioeconomic level and access to communication services, influence of the results. We have included in the study limitations section these considerations.
- I did not find the limits of the study specified
We have included a paragraph commenting on the limits of this study at the end of the discussion section.
- What are the criteria for excluding respondents?
Questionnaires with incomplete information or duplicates sent by the same email account were discarded. In the latter case, the first questionnaire received was chosen as long as it was complete or, failing that, subsequent ones were chosen with the same criteria.
We have added this information to the end of the second paragraph in the section Materials and Methods/Study population and Data collection.
- What is the reason why you introduced COVID-19 as the independent variable? How does it influence the results?
We included the Covid-19 variable to find out if it influenced the condition of suffering from any symptoms of periodontal disease related to the social development index, however, in the implemented models we did not find a significant association association.
- The references chapter does not fully respect the recommended style
Thanks for the comment, we reworked the references style.
- In my opinion, the citation index is much too high and should be revised. (I attach the Semplag report)
Thank you for your observation. We have extensively rewritten the whole manuscript, in particular the introduction and discussion sections. After doing this, we used the services of a professional English speaking copy-editor and proofreader to improve the readability of the manuscript.
- Are you thinking of continuing your research and if so, in what way?
Yes, thanks for your question. This work is indeed part of a broader research framework. On it, we are interested in studying the associations of periodontal disease with other diseases such as hypertension (see, for instance, Martínez-García, M., Castrejón-Pérez, R. C., Rodríguez-Hernández, A. P., Sandoval-Motta, S., Vallejo, M., Borges-Yáñez, S. A., & Hernández-Lemus, E. (2022). Incidence of arterial hypertension in people with periodontitis and characterization of the oral and subgingival microbiome: A study protocol. Frontiers in Cardiovascular Medicine, 8, 1929.) and diabetes, as well as the association of PD with social determinants of health and risk factors, and its relationships with chronic inflammation and immunological factors (work in progress). Regarding this particular branch project, our following steps would be to validate the self-report periodontal symptoms via clinical exploration and measurement sessions.

Round 2
Reviewer 3 Report
The authors modified the manuscript in line with my suggestions. The authors should improve the abstract by focusing the methods on a few points that are the keystones of the paper, such as the SDI, QASH, and bone loss. The authors added a detailed explanation of the SDI to the introduction. Anyway, the introduction should contain a brief explanation, whereas the methods should show the details. I suggest explaining the reason why the positivity to COVID-19 matters for the manuscript.
In my opinion, the study should be considered for publication after MAJOR REVISIONS.
With warm regards,
Author Response
The authors want to thank Reviewer 3 for their thorough review of our work. In what follows, we will present our response to your comments and suggestions in bold type.
The authors modified the manuscript in line with my suggestions. The authors should improve the abstract by focusing the methods on a few points that are the keystones of the paper, such as the SDI, QASH, and bone loss.
We have modified the abstract according to your suggestions. Simplified some off-topic comments and focused the abstract highlighting the role of SDI, QASH, and bone loss.
The authors added a detailed explanation of the SDI to the introduction. Anyway, the introduction should contain a brief explanation, whereas the methods should show the details. I suggest explaining the reason why the positivity to COVID-19 matters for the manuscript.
We agree with Reviewer 3. A brief mention of SDI in the introduction and a full technical description in the Methods section makes much more sense and greatly improves readability and clarity. Thank you!
In my opinion, the study should be considered for publication after MAJOR REVISIONS.
With warm regards,
We have modified our manuscript by closely following all your suggestions. A manuscript with tracked changes is included here.

Reviewer 4 Report
I went through the revised form of the manuscript, I believe that the authors have corrected the requests.
Author Response
The authors want to thank Reviewer 4 for their professional assessment of our work.
I went through the revised form of the manuscript, I believe that the authors have corrected the requests.
Thank you for reviewing our revised manuscript.